# Bioinformatics Analysis of RNA-seq Data Reveals Genes Related to Cancer Stem Cells in Colorectal Cancerogenesis

**DOI:** 10.3390/ijms232113252

**Published:** 2022-10-31

**Authors:** Kristian Urh, Nina Zidar, Emanuela Boštjančič

**Affiliations:** Institute of Pathology, Faculty of Medicine, University of Ljubljana, 1000 Ljubljana, Slovenia

**Keywords:** colorectal cancer, bioinformatics analysis, RNA-seq, cancer stem cells, differentially expressed genes, adenoma-carcinoma progression, metastases, qPCR

## Abstract

Cancer stem cells (CSC) play one of the crucial roles in the pathogenesis of various cancers, including colorectal cancer (CRC). Although great efforts have been made regarding our understanding of the cancerogenesis of CRC, CSC involvement in CRC development is still poorly understood. Using bioinformatics and RNA-seq data of normal mucosa, colorectal adenoma, and carcinoma (*n* = 106) from GEO and TCGA, we identified candidate CSC genes and analyzed pathway enrichment analysis (PEI) and protein–protein interaction analysis (PPI). Identified CSC-related genes were validated using qPCR and tissue samples from 47 patients with adenoma, adenoma with early carcinoma, and carcinoma without and with lymph node metastasis and were compared to normal mucosa. Six CSC-related genes were identified: *ANLN*, *CDK1*, *ECT2*, *PDGFD*, *TNC*, and *TNXB*. *ANLN*, *CDK1, ECT2,* and *TNC* were differentially expressed between adenoma and adenoma with early carcinoma. *TNC* was differentially expressed in CRC without lymph node metastases whereas *ANLN*, *CDK1,* and *PDGFD* were differentially expressed in CRC with lymph node metastases compared to normal mucosa. *ANLN* and *PDGFD* were differentially expressed between carcinoma without and with lymph node metastasis. Our study identified and validated CSC-related genes that might be involved in early stages of CRC development *(ANLN, CDK1, ECT2, TNC)* and in development of metastasis *(ANLN, PDGFD).*

## 1. Introduction

A majority of any given neoplasm presumably consists of cells incapable of tumor progression and formation of metastases. It is believed that for tumor growth and metastatic spread, a minority of cancer cells is needed, referred to as cancer stem cells (CSC) or CSC-like cells, which are capable of self-renewal, differentiation, and mobility. Numerous studies suggest that CSC have a higher propensity toward invasion and that they are also responsible for disease relapse [1]. One reason might be due to fact that turnover of CSCs is slow, which in turn allows greater resistance to therapies that target rapidly replicating cells [2,3].

Our previous studies showed that genes related to CSC-like cells might be also involved in development and progression of colorectal cancer (CRC) [4,5]. CRC is one of the most common causes of morbidity due to cancer worldwide [6]. For patients with CRC, the five-year survival rate varies, approximately 90% in patients with early lesions and 8–12% in patients with advanced CRC [7]. Despite new treatments options, 40–50% of CRC patients develop metastases [6,7,8,9]. The prognosis is significantly improved with early lesion detection through population screening programs [10,11]. CRC progression from normal mucosa to invasive carcinoma mostly develop through precursor lesions, i.e., adenomas [9]. Molecular pathways involved in CRC development are widely recognized, e.g., chromosomal instability (CIN) pathway, microsatellite instability (MSI) pathway and epigenetic instability pathway (CpG island methylator phenotype, CIMP) [9,12]. However, most genetic events that are associated with tumor development occur early, before the formation of the carcinoma, consequently leading to a need to define mechanisms and biomarkers responsible for malignant transformation.

Genes associated with CSC features might thus be potentially promising biomarkers in CRC development and progression. In CRC patients, CSC-associated molecular profiles can predict tumor regeneration and disease relapse after conventional therapy [3,13,14,15,16,17]. Moreover, several potential anti-CSC targeted drugs have been identified, with some of them making their way to the clinic [18]. However, knowledge about CSC involvement in early stages of CRC development, as CRC progresses from adenoma to early carcinoma is limited. Up to date, several CSC-related markers have been described in adenomas, e.g., *CD133* [19], *LGR5*, *PROM1* [20], *IL-8* [21], *OLFM4*, and *ASCL2* [22], but more information is required to understand malignant transformation, despite current findings in the field [19,20,21,22,23].

By comparing transcriptional heterogeneity in primary CRC and corresponding metastases, fewer differences were seen between the primary tumors and their matched metastases in comparison to marked differences in protein-coding gene expression profiling between the normal mucosa and primary CRC. These results suggest that the vast majority of gene expression changes manifest in the early stages of the CRC development [24,25] and support the necessity to investigate precancerous lesions and early CRC. There are numerous genome wide and high-throughput analyses investigating expression and/or methylation status of CRC in comparison to the normal mucosa [22,26,27,28,29,30,31,32,33]. Publications investigating colorectal adenoma to adenocarcinoma and normal mucosa were in the past mainly focused on proteomic analyses and more recently mainly on the serum and plasma expression profiling from CRC patients [29,34,35]. However, analyses of gene expression investigating adenoma in comparison to adenocarcinoma are limited and have thus far not elucidated the CSC involvement in CRC development and progression [20,22,23,26,36,37,38,39,40,41,42,43,44].

The aim of our study was to identify differentially expressed genes (DEGs) related to CSC in CRC development through bioinformatics analysis of raw RNA-seq projects data containing samples of normal mucosa, adenomas, and carcinomas. Identified genes were chosen for further validation with qPCR on our own cohort of samples for their involvement in CRC development and progression.

## 2. Results

### 2.1. Identification of DEGs

#### 2.1.1. Identification of DEGs from Gene Expression Omnibus (GEO) Database

Identification of DEGs is summarized in Table 1. All procedures for identification of DEGs were performed at specified cut-off conditions. The full results, including the pathway enrichment analysis (PEI) and protein–protein interaction analysis (PPI), are available in Appendix A.

#### 2.1.2. Comparing Results between Projects from GEO Database—CRC and Corresponding Normal Mucosa

In the first comparison, we included experiments performed on the Ion Torrent sequencer. We identified 384 DEGs between all three projects that included normal tissue and carcinoma pairs, GSE50760, GSE104836, and GSE100243. Project GSE92914 was excluded from this analysis, as there were no DEGs to compare. The full list of overlapping DEGs is available in Appendix A, sheet 1. Venn diagram showing the overlapping DEGs between projects is shown in Figure 1a.

***PEI analysis.*** Through the gene ontology (GO) analysis on the list of overlapping genes, we identified 48 biological processes, 14 cellular localization terms, and 15 molecular functions. The full results are available in Appendix A, sheet GO. We identified 6 pathways associated with the overlapping DEGs in the KEGG database and 17 pathways in the REACTOME database using the DAVID tool and previously mentioned settings. The full results are available in Appendix A, sheet Path.

***PPI analysis.*** The PPI analysis on the overlapping DEGs resulted in 32 STRING local network terms shown in Appendix A, sheet String.

The identified predicted functional partners are *TRIP13*, *PSMD8*, *AURKA*, *NCAPG*, and *CCNB1*.

In the next comparison, we performed overlaps between projects, including carcinoma and corresponding normal mucosa, but excluding experiments performed on the Ion Torrent sequencer. We identified 954 DEGs between two projects that included carcinoma and corresponding normal mucosa, GSE50760 and GSE104836 that were performed on the same sequencing technology. Project GSE92914 was excluded from this analysis as there were no DEGs to compare. The full list of overlapping DEGs is available in Appendix A, sheet 1. Venn diagram showing the overlapping DEGs between projects is shown in Figure 1b.

***PEI analysis.*** Through the GO analysis on the list of overlapping genes, we identified 88 biological processes, 24 cellular localization terms, and 23 molecular functions. The full results are available in Appendix A, sheet GO. We identified 17 pathways associated with the overlapping DEGs in the KEGG database and 33 pathways in the REACTOME database using the DAVID tool and previously mentioned settings. The full results are available in Appendix A, sheet Path.

***PPI analysis.*** The PPI analysis on the overlapping DEGs resulted in 52 STRING local network terms shown in Appendix A, sheet String.

The predicted functional partners are *PSMD8*, *KAT2A*, *GNB3*, *GNB4,* and *ORC2*.

#### 2.1.3. Identification of DEGs through Joint Projects—CRC and Adenomas

For this analysis, we joined the counts data from projects GSE72820, GSE104836, GSE50760, GSE92914, and GSE76987 performed on the same sequencing technology (Illumina) in one project and analyzed the data.

For comparisons between normal tissue and carcinomas at logarithm of base 2 of the fold change value (logFC) > 1.5 and adjusted *p* < 0.05, we identified 1524 DEGs. The full list of DEGs is available in Appendix A, table NvsC, sheet 1. Additionally, the top 100 genes from this analysis are featured in the heatmap shown in Figure 2. For comparisons between adenomas and carcinomas at logFC > 1.5 and adjusted *p* < 0.05, we identified 706 DEGs. The full list of identified DEGs is available in Appendix A, table AvsC, sheet 1. Additionally, the top 100 genes from this analysis are featured in the heatmap shown in Figure 3.

When overlapping 706 DEGs between adenomas and carcinomas with 1524 DEGs between normal tissue and carcinomas, we identified 569 overlapping DEGs. The full list of overlapping DEGs and the PEI and PPI analyses is available in Appendix A, table ACvsNC, sheet 1.

***PEI analysis.*** Through the GO analysis on the list of overlapping genes, we identified 67 biological processes, 16 cellular localization terms, and 21 molecular functions. The full results are available in Appendix A, table ACvsNC, sheet GO. We identified 17 pathways associated with the overlapping DEGs in the KEGG database and 18 pathways in the REACTOME database using the DAVID tool and previously mentioned settings. The full results are available in Appendix A, table ACvsNC, sheet Path.

***PPI analysis.*** The PPI analysis on the overlapping DEGs resulted in 36 STRING local network terms shown fully in Appendix A, table ACvsNC, sheet String. The full PPI network is shown in Figure 4a, while the physical PPI network is shown in Figure 4b. The physical PPI network represents shared physical complexes. The edges indicate that the directly linked proteins are part of the same physical complex.

The predicted functional partners are *AURKA*, *GNB3*, *GNB4*, *ORC2*, and *NCAPG*.

#### 2.1.4. TCGA Analysis

The analysis of data from The Cancer Genome Atlas (TCGA) database was performed on 519 samples from TCGA-COAD (colon) RNA-seq project available for public analysis. The HTSeq-Counts workflow data were obtained for primary tumors and normal tissue samples. The independent analysis between groups was performed as explained in the Materials and Methods section.

The aforementioned procedure for identification of DEGs in this project resulted in 4259 DEGs at logFC > 1.5 and *p* < 0.05, after adjustment, for comparing normal tissue and carcinomas. Top 100 DEGs are shown in Figure 5 heatmap of expression between the primary tumor and normal samples. The full results including PEI analysis is available in Appendix A, table TCGA. The number of DEGs was too high to perform PPI analysis. PPI analysis and PEI analysis were also performed separately for the upregulated and downregulated (only the first 2000 per STRING limit) DEGs. The results are available in Appendix A, table TCGA upregulated, table TCGA downregulated.

#### 2.1.5. Comparing Results from GEO Projects and TCGA Results

The 1524 DEGs identified between the normal tissue and carcinoma tissue from GEO (Appendix A, table NvsC, sheet 1) were compared to the 4259 DEGs identified in the TCGA analysis. We identified 1165 overlapping DEGs. The full list of overlapping DEGs is available in Appendix A, sheet 1.

***PEI analysis.*** Through the GO analysis on the list of overlapping genes, we identified 132 biological processes, 38 cellular localization terms, and 41 molecular functions. The full results are available in Appendix A, sheet GO. We identified 16 pathways associated with the overlapping DEGs in the KEGG database and 40 pathways in the REACTOME database using the DAVID tool and previously mentioned settings. The full results are available in Appendix A, Path.

***PPI analysis.*** The PPI analysis on the overlapping DEGs resulted in 44 STRING local network terms shown fully in Appendix A, sheet String.

The predicted functional partners are *RCF2*, *MCM5*, *AURKA*, *ORC2,* and *NCAPG*.

### 2.2. Identification of Candidate Genes for Further Validation

Among the identified DEGs in the joint GEO projects analysis, also the DEGs identified comparing adenoma and corresponding normal mucosa as well as overlap between carcinomas and corresponding normal tissue were considered for further validation. In the PEI analysis from the adenoma versus carcinoma comparison, the PI3K-Akt signaling pathway, associated with CSC [45,46,47] was chosen for identification of candidates for further validation. In the PPI analysis CL:12302 mixed, incl. mitotic cytokinesis, and G2/M DNA replication checkpoint and CL:12305 mixed, incl. mitotic cytokinesis, and G2/M DNA replication checkpoint terms were chosen for further validation. The DEGs from the aforementioned pathway and from the PPI analysis were then checked for references associating them with CSC or stem cell terms and checked for presence in the Stemformatics database [48]. Finally, six genes were identified for further validation, namely anillin, actin binding protein (*ANLN*), cyclin dependent kinase 1 (*CDK1*), epithelial cell transforming 2 (*ECT2*), platelet derived growth factor D *(PDGFD*), tenascin C (*TNC*) and tenascin XB (*TNXB*). The summary of comparisons is represented in Figure 6.

### 2.3. Validation of Identified Candidate Genes on Samples of Normal Mucosa, Adenoma, Adenoma with Early Carcinoma, and Carcinoma without and with Lymph Node Invasion

#### 2.3.1. Differential Gene Expression in Adenoma and Adenoma with Early Carcinoma

∆Cq for the investigated genes in adenoma and adenoma with early carcinoma were statistically evaluated independently against normal mucosa samples. Statistically significant results in adenomas include 12.4-fold downregulation of *TNC* (*p* ≤ 0.001), 9.4-fold downregulation of *TNXB* (*p* ≤ 0.001), and 6.0-fold upregulation of *ECT2* (*p* ≤ 0.001). In adenomas with early carcinomas, statistically significant results include *TNXB* with 7.7-fold downregulation (*p* ≤ 0.001), 4.4-fold upregulation of *CDK1* (*p* ≤ 0.01), 14.1-fold upregulation of *ANLN* (*p* ≤ 0.001), and 26.3-fold upregulation of *ECT2* (*p* ≤ 0.001). We identified statistically significant differences between the adenoma and adenoma with early carcinoma for *ANLN* (*p* ≤ 0.001), *CDK1* (*p* ≤ 0.05), *ECT2* (*p* ≤ 0.01), and *TNC* (*p* ≤ 0.05). The results are shown in Figure 7a.

#### 2.3.2. Gene Expression in Carcinoma Compared to Corresponding Normal Mucosa

∆Cq for the investigated genes in CRC without and with lymph node metastases were statistically evaluated against corresponding normal mucosa. In CRC without lymph node metastases, 4.8-fold upregulation of *TNC* (*p* ≤ 0.05) was observed. The results are shown in Figure 7b.

In CRC with lymph node metastases, there was 51.4-fold upregulation of *PDGFD* (*p* ≤ 0.01), 2.7-fold upregulation of *CKD1* (*p* ≤ 0.05), and 7.9-fold upregulation (*p* ≤ 0.05) of *ANLN.* The results are shown in Figure 7c.

When comparing the ∆∆Cq data (normalized to paired mucosa samples) of CRC without to CRC with lymph node metastases, we identified statistically significant upregulation at *p* ≤ 0.05 for *PDGFD* and *ANLN*. Results are shown in Figure 7d.

We report all of the investigated comparisons in the tested groups below in Table 2.

#### 2.3.3. Association with Level of Malignancy

Additionally, we identified positive correlation of several genes in relation to malignancy levels (from normal mucosa, adenoma, adenoma with early carcinoma, CRC without to CRC with lymph node metastasis) with the use of Spearman correlation coefficient. The correlated genes were *PDGFD* (r_s_ = 0.637; *p* ≤ 0.001)*, CDK1* (r_s_ = 0.256; *p* ≤ 0.05)*, ANLN* (r_s_ = 0.453; *p* ≤ 0.001), and *ECT2* (r_s_ = 0.312; *p* ≤ 0.05).

## 3. Discussion

In the present study, we identified six CSC-related genes associated with CRC development by analyzing RNA-seq data from GEO and TCGA databases, comparing data of normal mucosa, adenoma, and CRC. Previous bioinformatics studies on CRC have identified many gene expression profiles mainly between normal tissue, carcinoma, and metastases; however, studies on gene expression profiles between adenoma and carcinoma are less common [22,23,26,36,37,38,39,40,41,42,43,44,49,50,51,52,53]. The DEGs between adenoma and carcinoma are important for identification of potential biomarkers of malignant transformation [22,23,38,39,40,42,43,44,51,52,53], some of which were previously identified by our group, by analyzing publicly available data on microarray analyses from the GEO database [26]. Whereas the previous bioinformatics studies were focused on any differential expressed gene between adenoma and CRC, in the current study, we focused on genes that are related to CSC. We performed an independent analysis irrespective of original procedures. The six identified CSC-related genes could be potential biomarkers indicating malignant potential and cancer progression.

### 3.1. Identified DEGs, PEI and PPI Related to CSC

By combining results of carcinoma compared to normal tissue and normalized adenoma and carcinoma compared to each other, we identified 384 and 706 overlapping DEGs, respectively. We identified several pathways that can be associated with CSC alone or both CSC and extracellular matrix (ECM). Some of them were identified by our group in previous studies, associated with the ECM and CSC, and were already validated in CRC development [4,54]. The identified pathways in current study were the ECM-receptor interaction pathway [55], PPAR signaling pathway [56], and the PI3K-Akt signaling pathway [45,46,47]. Six genes were chosen for further validation after identification of association with CSC: three DEGs were associated with CSC, namely platelet derived growth factor D *(PDGFD*), tenascin C (*TNC*) and tenascin XB (*TNXB*) from the pathway PI3K-Akt, and three DEGs, namely anillin actin binding protein (*ANLN*), cyclin dependent kinase 1 (*CDK1*), and epithelial cell transforming 2 (*ECT2*) that are associated with G2/M DNA replication checkpoint in the PPI analysis.

Previous studies suggested that *TNC* positively correlates with *SOX2* and AKT/HIF1a and with stem cell properties [57]. *TNXB* induced invasive growth and metastases in tumor cells with stem cell properties [58]. Its expression was downregulated during cancer progression, while upregulation was associated with better prognosis [59]. *PDGFD* was reported to be associated with epithelial-mesenchymal transition (EMT) and stem cells in a breast cancer cell line, where it was also associated with tumor growth [60]. Its expression in mesenchymal stem cells obtained from stomach carcinoma was associated with cancer progression in vitro and in vivo [61]. *CDK1* was associated with maintenance of stem cell properties in lung cancer by interacting with *SOX2* [62], another stem cell marker in CRC, and its protein expression increases significantly from normal mucosa to dysplasia and from dysplasia to carcinoma [63,64]. It was also associated in maintenance of pluripotency and genomic stability in human pluripotent stem cells [65]. *ANLN* was associated with migration, growth, and metastasis of breast cancer cells through control of stemness and differentiation [66]. It also enhanced stemness of triple negative breast cancer [67]. Higher expression of *ECT2* was associated with worse prognosis in stomach cancer, where its transcriptomic profile was associated with stem cell properties in cancer cells and was suggested as a potential biomarker for cancer cells with stem cell properties [68].

### 3.2. Expression of Investigated Genes in Association with Malignant Transformation, Progression, and Mestastases of CRC

The genes *TNC*, *CDK1*, *ANLN,* and *ECT2* were significantly upregulated in adenoma with early carcinoma when compared to adenoma and can be considered as potential biomarkers for malignant transformation. To the best of our knowledge, no relevant data on *ANLN* expression in adenoma in comparison to carcinoma of CRC has been published so far. *CDK1* was identified as expressed in adenomas and CRC, while expression of its partner *CDKN1A* was absent, suggesting that the proliferative activity of CRC, which may be an early event in carcinogenesis, is not solely dependent on control of the cell cycle by these two genes [69]. Similarly, we identified *CDK1* upregulation in adenomas with early carcinomas and when compared to adenomas. *ECT2* was identified as overexpressed in adenomas and CRC, elevated nucleated *ECT2* correlated with poorly differentiated tumors, a low cytoplasmic to nuclear ratio of ECT2 was associated with poor patient survival and suggests that this gene is a driver of tumor progression [70]. We similarly identified its upregulation from normal mucosa to adenomas and adenomas with early carcinoma. The expression of *TNC* was identified in adenomas and CRC, often in invading edges, it was associated with a positive relationship between degree of protein-coding gene expression levels and the depth of invasion, as well as frequency of metastasis, suggesting an important role during cancer development and progression [71]. Although we identified downregulation of *TNC* in adenomas when compared to normal mucosa, we observed upregulation when comparing adenoma with early carcinoma to adenoma.

The gene *TNC* was upregulated in CRC without lymph node metastases when compared to corresponding normal mucosa. *ANLN*, *CDK1,* and *PDGFD* were upregulated in CRC with lymph node metastases compared to corresponding normal mucosa. *ANLN* and *PDGFD* were significantly upregulated in CRC with lymph node metastases when compared to those without lymph node metastases and can be considered as potential biomarkers for metastases in CRC.

For some genes, information is available in the field of cancer progression and metastases development. Expression of TNC protein in stromal cells of CRC was associated with late-stage CRC and a shorter long-term survival [72], similarly, in our research it was upregulated in CRC without lymph node metastases. *PDGFD* downregulation was associated with lower proliferation levels not only in CRC but also in prostate cancer [73]. Its upregulation was also associated with cancer cell migration, invasiveness, and proliferation in CRC [74]. Similarly, in our research it was significantly upregulated in CRC with lymph node metastases compared to CRC without lymph node metastases and corresponding normal mucosa. It also positively correlated with the level of malignancy. Upregulation of *ANLN* was associated with CRC progression and poor prognosis [75]. Similarly to *PDGFD,* in our research, *ANLN* was upregulated in CRC with lymph node metastases when compared to normal mucosa and to CRC without lymph node metastases and positively correlated with the level of malignancy. It is regulated by *SP2* and promotes proliferation of CRC cells through the PI3K/AKT and MAPK pathways [76]. High nuclear/cytoplasmic ratio of *CDK1* expression predicted poor prognosis in CRC patients [77]. Similarly, in our research it was upregulated in CRC with lymph node metastases and was positively correlated with the level of malignancy. In CRC patients harboring *BRAF* mutations, apoptosis resistance can be overcome when the therapeutic targets are *CDK1* and *MEK*/*ERK* [78]. Upregulation of *ECT2* was associated with a shorter survival without CRC recurrence [79] and a shorter overall survival in CRC patients [79,80]. While we identified no significant differences in expression of *ECT2* between CRC and their corresponding normal mucosa or between CRC without and with lymph node metastases, our findings of positive correlation with malignancy levels are in line with the findings in the field.

Additionally, through identification of positive correlation of *PDGFD*, *ANLN*, *CDK1,* and *ECT2* to the level of malignancy, we propose these genes to be considered also as potential biomarkers for CRC progression.

### 3.3. The Most Common Predicted Functional Partners in the PPI Analysis

The most common predicted functional partners in the PPI analysis of our CRC data were *AURKA, ORC2,* and *NCAPG*.

*AURKA* plays important roles in mitosis and chromosome stability, it is a prognostic marker in CRC [81], it increased the chemosensitivity of colon cancer cells to oxaliplatin by inhibiting the TP53-mediated DNA damage response genes [82], and its expression in CRC liver metastases was associated with poor prognosis [83]. Additionally, it promoted cancer metastasis by regulation of EMT and CSC properties in hepatocellular carcinoma [84], and it enhanced the breast CSC phenotype [85].

Depletion of *ORC2* was associated with mitotic arrest due to chromosome condensation defects. In hepatocellular carcinoma, *ORC2* was not associated with overall survival; however, its expression was correlated with disease-free survival [86]. It was also upregulated in microarray data analysis on the cisplatin-sensitive T24 and the cisplatin-resistant T24R2 bladder cancer cell line [87]. It was identified that HCT116 colon cancer cells initiate DNA replication normally in the absence of *ORC5* and *ORC2* proteins [88]. Additionally, *ORC2* was associated with prevention of apoptosis induction and could serve as a potential target for cancer therapy by inhibition of cancer cell stemness [89].

*NCAPG* facilitated colorectal cancer cell proliferation, migration, invasion, and EMT by activation of the Wnt/β-catenin signaling pathway [90]. Additionally, *miR-23b-3p*/*NCAPG*/*PI3K*/*AKT* signaling axis may play an important role in CRC carcinogenesis [91].

Through validation of expression of the investigated genes in normal mucosa, adenomas, adenomas with early carcinomas, and CRC without and CRC with lymph node metastases, we identified potential CSC-related biomarkers for malignant transformation, progression, and metastases of CRC. Our results are further supported by PPI analyses and most common predicted functional partners.

### 3.4. Limitations of the Study

An important limitation of our study is associated with the use of normal mucosa samples. Although they were taken at least 20 cm away from the tumor and showed no microscopic abnormalities, genetic and protein aberrations may already be present in morphologically normal mucosa [92,93]. However, these samples can still be used as corresponding control samples to overcome differences in the physiological and genetic background. Another limitation of our study is a relatively small sample size. The latter is due to the use of formalin-fixed paraffin-embedded (FFPE) tissue samples, in which nucleic acids are challenging to analyze. Therefore, only samples that successfully passed the initial quality control and samples with stable expression of the reference genes were selected for further analysis, thus limiting the number of included samples. Moreover, further validation on a higher number of samples may be needed for additional confirmation of potential biomarkers.

## 4. Materials and Methods

### 4.1. Bioinformatics Analysis of Publicly Available Data

#### 4.1.1. Project Selection

Several projects (GSE72820, GSE104836 [50], GSE50760, GSE92914 [49], GSE76987 [51], GSE100243) with gene expression profiles of colon normal, and/or adenoma, and/or carcinoma were downloaded from the public functional genomics data repository Gene Expression Omnibus database (GEO, http://www.ncbi.nlm.nih.gov/geo) (accessed on 30 March 2020) of the National Center for Biotechnology Information (NCBI). Where available, TNM status was considered for project inclusion. Only samples where corresponding normal tissue was available were included in the analysis. Five projects were performed on Illumina HiSeq/NextSeq technology, while one (GSE100243) was performed on Ion Torrent Proton technology. An overview of projects included samples, and additional information is shown in Table 3.

We also included data downloaded from the cancer genome atlas, TCGA-COAD data, which included in total 519 samples, 478 primary tumors, and 41 normal tissue samples.

#### 4.1.2. Data Processing

For all GEO projects, the original raw data fastq files were downloaded from ENA (https://www.ebi.ac.uk/ena/browser/home) (accessed on 30 March 2020). All projects, including TCGA data, were processed in R language (https://www.r-project.org/) (accessed on 30 March 2020).

First the fastq, data were checked using the fastQC tool (Babraham Bioinformatics—FastQC A Quality Control tool for High Throughput Sequence Data). The full index for the reference genome »GRCh38« was first built using the function »buildindex« of the Rsubread package [94]. The fastq files were aligned to the reference GRCh38 genome using the align function of Rsubread package. Additional trimming of the reads was not performed as the Rsubread aligner performs “soft clipping” according to publication [95]. Afterward, mapped sequencing reads were assigned to specified genomic features using the function »featureCounts« of Rsubread package. Annotation was performed on human reference genome GRCh38. The feature counting summary file was checked for each sample using the FastQC tool.

#### 4.1.3. Analysis of Projects Obtained from GEO/TCGA Database

Data from selected GEO projects and data from TCGA were analyzed separately.

For data from GEO projects, samples from each project were merged into one corresponding object using the BiocGenerics package [96]; tissue group information for each sample was obtained from GEO project page or by obtaining the series matrix file from GEO using the GEOquery package [97]. Annotation data for genes and gene length was obtained from a feature counted sample file. Counts for each sample, tissue group info, and annotation data were joined using EdgeR package [98].

TCGA count data were first preprocessed using the TCGAbiolinks package. Afterward, further data processing was performed as for GEO projects.

Count data were filtered by using the edgeR package. After filtration, the first round of quality control was performed on count per million data (cpm) and log2 count per million data (lcpm). Density plots and boxplots were compared for unfiltered and filtered data. A dendrogram was constructed, and distances were calculated using lcpm data with Spearman method. A MDS plot was also constructed from the distances.

Afterward, gene expression distribution was normalized across samples using the method trimmed mean of M values (TMM). A new round of quality control using cpm and lcpm data was performed on the normalized project by density plots, box plots, dendrogram, and the MDS plot. Design of the matrix was performed as paired samples for GEO data and as independent samples for TCGA data. The design matrix was afterward used in the voom normalization step and linear modeling. Before linear modeling, the »voom« function from the Limma package [99] was used for transforming the filtered and normalized count data to logCPM and estimation of the mean-variance relationship, using these estimates to compute appropriate observation-level weights.

The processed data were fitted with a linear model for each gene. For the independent comparisons, contrasts.fit of the Limma package was used to compute contrasts from linear model fit and the previously designed contrasts matrix. The eBayes function of the Limma package was used to compute moderated t-statistics, moderated F-statistic, and log-odds of differential expression from a given linear model fit. Finally, the topTable function of the Limma package was used to extract a table of the top-ranked genes from a linear model fit with a *p*-value less than 0.05 and an absolute logFC more than 1.5 with the Benjamini Hochberg method of adjusting *p*-value.

### 4.2. Pathway Enrichment Analysis of Publicly Available Data

Pathway enrichment analysis was performed by using Database for Annotation, Visualization and Integrated Discovery (DAVID) v6.8 [100]. Entrez gene IDs were uploaded to the service and homo sapiens was selected as the species. We analyzed the GO results for biological process, cellular component, and molecular function data at the following settings: the count cutoff was at 2, the Modified Fisher Exact *p*-value (EASE) at 0.01, and Bonferroni statistic was chosen at the display level. Pathway results were analyzed for Biocarta, if available, Kyoto Encyclopedia of Genes and Genomes (KEGG) and REACTOME. Settings used in this analysis were the same as with GO results. Results of both analyses were saved to a corresponding excel file with the analyzed DEGs.

### 4.3. Protein–Protein Interaction Analysis

For construction of the protein–protein interactions (PPI) network, the Search Tool for the Retrieval of Interacting Genes (STRING) database was used (https://string-db.org/) (accessed on 30 March 2020) [101]. PPI network analysis is an important tool for interpretation of molecular mechanisms in the process of carcinogenesis. The settings used in the analyses were a required score at the highest confidence (0.900), a FDR stringency at medium setting (5 percent); a meaning of network edges—confidence; textmining was removed from the active interaction sources; disconnected nodes were not shown in the network; and only query proteins were considered for the interactions. Additionally, the first analysis did not include clustering; afterward, the k-means clustering and Markov cluster algorithm (MCL) clustering methods were applied separately, with the number of clusters set at the default 3. We also performed the analysis with the physical network chosen as the edges indicate that the proteins are part of a physical complex. Lastly, we allowed for the identification of predicted functional partners with the setting of »no more than 5 interactors« on the first shell. All networks were exported as a bitmap image corresponding to the specific DEGs analysis, and the local network clustering analysis from STRING was considered.

### 4.4. Validation of Selected Potential Candidate Genes

#### 4.4.1. Patient and Tissue Selection

Patients who underwent excision or resection of adenoma, adenoma with early carcinoma, and CRC, from 2015 to 2019, were included in the study. For routine histopathologic examination, tissue samples were fixed in 10% buffered formalin and embedded in paraffin (FFPE). During routine examination, all specimens were evaluated by pathologists according to standard procedures and, after histopathologic examination, pTNM (pathologic tumor node metastasis) classification was assessed on the basis of the depth of invasion and the extent of the primary tumor, the number of lymph nodes with metastases, and the presence of distant metastases. For the purpose of this study, biopsy samples were collected retrospectively from the archives of the Institute of Pathology, Faculty of Medicine, University of Ljubljana. After reevaluation of consecutive cases for each group by a pathologist and initial quality check, representative samples were selected for further study. Samples of normal mucosa obtained from resected CRC specimens were used as control samples. Patients treated by radiotherapy, chemotherapy or biologic drugs prior to surgery were not included in this study. Patients with mucinous carcinomas or signet cell carcinomas were also excluded. Only sporadic CRC cases were included. Tissue samples were grouped as normal mucosa, adenoma, adenoma with early carcinoma, CRC without lymph node metastases (CRC N0), or CRC with lymph node metastases (CRC N+).

The study was conducted according to the guidelines of the Declaration of Helsinki and approved by the National Medical Ethics Committee (Republic of Slovenia, Ministry of Health), approval number 0120-54/2020/4.

Approximately 30% of retrospectively selected cases successfully passed initial quality control, as described below (4.6.2). Our study therefore included 70 biopsy samples from 47 patients with adenoma (*n* = 11), adenoma with early carcinoma (*n* = 13), CRC without lymph node metastases (*n* = 10), and CRC with lymph node metastases (*n* = 13). There were 15 women and 32 men, aged 73.7 ± 8.4 and 65.7 ± 11.4 years, respectively. As a control group, microscopically normal mucosa from CRC resected specimens was used (*n* = 23). Characteristics of the included patients are shown in Table 4.

#### 4.4.2. RNA Isolation and Quality Assessment

RNA was obtained from FFPE tissue slides using a microtome (4 × 10 µm-thick slides). RNA was isolated using an AllPrep DNA/RNA FFPE kit (Qiagen, Hilden, Germany) according to the manufacturer’s protocol. Concentration and quality assessment of the isolated RNA was performed using a spectrophotometer ND-1000 or ND-One (Nanodrop, Thermo Fisher Scientific, Waltham, MA, USA) at wavelengths 260 nm and 280 nm. Prior to further analysis, RNA quality was tested using reverse transcription and amplification of GAPDH (Hs_GAPDH_vb.1_SG, 100 bp) by SybrGreen technology. Samples that did not amplify during this initial control step were excluded from further analysis.

#### 4.4.3. Reverse Transcription (RT) and Pre-Amplification

Reverse transcription (RT) of the isolated mRNA was performed using OneTaq^®®^ RT-PCR Kit (New England Biolabs, Ipswich, MA, USA) using a mix of random hexamers and oligo-dT primers according to the manufacturer’s protocol. We used 60 ng of RNA in the total 10 µL RT reaction and 1 µL of random hexamers and incubated for 5 min at 70 °C. Afterward, we added 5 µL of the Reaction mix and 1 µL of Enzyme mix to the reaction and incubated at 25 °C for 5 min, 42 °C for 1 h and 80 °C for 5 min.

Preamplification of the obtained cDNA was performed using the TaqMan^®^ Preamp Master Mix (Thermo Fisher Scientific, Waltham, MA, USA) according to the manufacturer’s instructions. For a 10 µL reaction, we added 5 µL of Pre-Amp Master Mix (2×), 2.5 µL of Pooled TaqMan^®^ Gene Expression probes (Thermo Fisher Scientific, Waltham, MA, USA) (0.2×, diluted in TE buffer), and 2.5 µL of cDNA. Incubation was performed at 95 °C for 10 min, 95 °C for 15 s, and 60 °C for 4 min.

#### 4.4.4. Selection of Primers and Probes

The TaqMan-based approach (Thermo Fisher scientific, Waltham, MA, USA) was used for the quantitative real-time PCR (qPCR) methodology. A predesigned mixture of primers and probes was used for expression analysis of DEGs relative to reference genes (RGs). The genes were selected as described above. Selected probes are shown in Table 5, with reference genes (RGs) presented in bold.

#### 4.4.5. Quantitative Real-Time PCR (qPCR)

Prior to qPCR amplification, efficiencies were determined in triplicate reactions for each probe and for each group of samples. The dilution series included 4-point dilutions ranging from 5-fold to 625-fold. A Rotor Gene Q (Qiagen, Hilden, Germany) machine was used for all qPCR analyses, and all 10 µL testing reactions were performed in duplicate. The cycling protocol was 50 °C for 2 min, 95 °C for 10 min, 40 cycles of 95 °C for 15 s, and 62 °C for 1 min. The reactions included 5.0 µL of the FastStart™ PCR Master mix (Roche Diagnostics, Basel, Switzerland) (2×), 0.5 µL of the TaqMan assay (predesigned mixture of primers and probes) (20×), and 4.5 µL of cDNA (60 ng of RNA reverse transcribed to cDNA and pre-amplified, as explained in Section 4.4.4; the cDNA was diluted 5-fold).

After efficiency correction, the obtained ΔCq (normalized Cq of analyzed mRNAs relative to geometric mean of RGs) were used for analysis of target gene. The fold difference in the expression was calculated against the normal mucosa samples group using the ΔΔCq method [102].

#### 4.4.6. Statistics

Differences in expression were compared between tumor and corresponding normal mucosa using ΔCq and the Willcoxon rank test (nonparametric test for dependent samples). For comparison of relative quantification of mRNAs between independent groups of samples (e.g., adenoma vs. normal mucosa), ΔCq and the Mann–Whitney U test were used (nonparametric test for independent group of samples). ΔΔCq and the Mann–Whitney U test were used for comparison between CRC without and CRC with lymph node metastases. Using the Spearman coefficient, we analyzed whether protein-coding gene expression levels were associated with cancerogenesis. All statistical analyses of experimental data were performed using SPSS version 27 (SPSS Inc., Chicago, IL, USA). Differences in expression between groups were considered significant at *p* ≤ 0.05.

## 5. Conclusions

In conclusion, we analyzed publicly available RNA-seq data from normal mucosa, adenomas, and carcinomas of colon from six projects from GEO and TCGA databases for identification of differentially expressed CSC-related genes. We identified six genes associated with CSC in CRC cancerogenesis: *ANLN*, *CDK1*, *ECT2*, *PDGFD*, *TNC,* and *TNXB*. We showed that genes *ANLN*, *CDK1*, *ECT2,* and *TNC* were associated with adenoma progression to carcinoma, *ANLN* and *PDGFD* with development of metastases, while genes *ANLN*, *CDK1*, *ECT2,* and *PDGFD* were associated with the level of malignancy. Our results provide further evidence for the involvement of the CSC-associated genes in the development and progression of CRC, and several of them are promising candidates for further validation. Future perspectives include further validation on a higher number of samples for additional confirmation of the results and using immunohistochemistry to investigate localized expression of the tested genes to directly confirm these genes as potential biomarkers for CSC in CRC development.

## Figures and Tables

**Figure 1 ijms-23-13252-f001:**
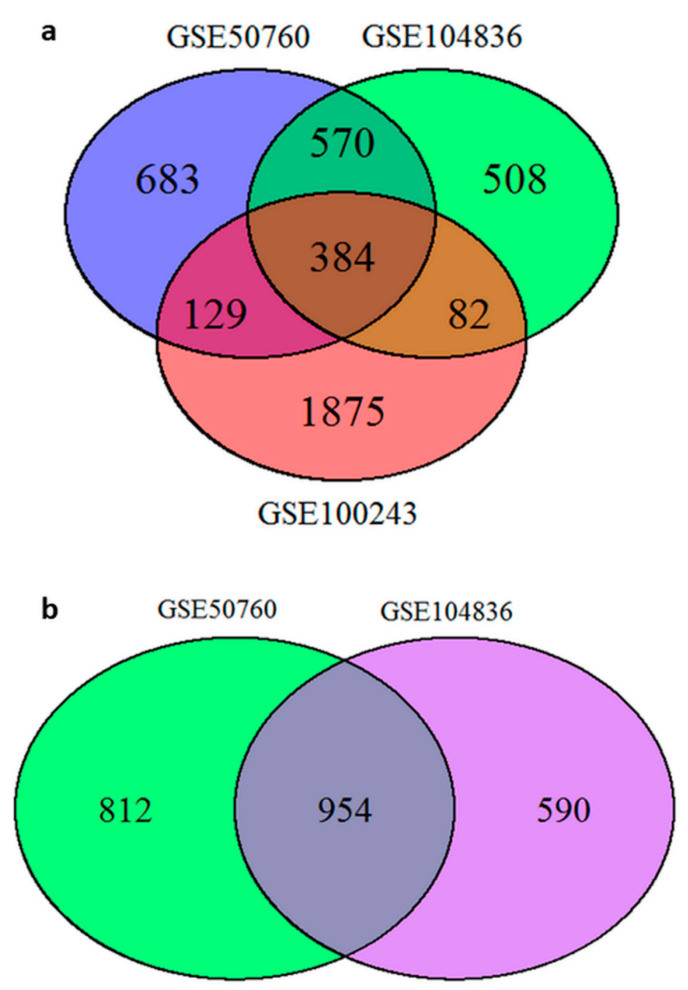
Venn diagrams of overlapping differentially expressed genes: (**a**) overlap between projects GSE50760 (blue color), GSE104836 (green color), and GSE100243 (red color); (**b**) overlap between project GSE50760 (green color) and GSE104836 (violet color). Legend: GSE: gene expression omnibus series.

**Figure 2 ijms-23-13252-f002:**
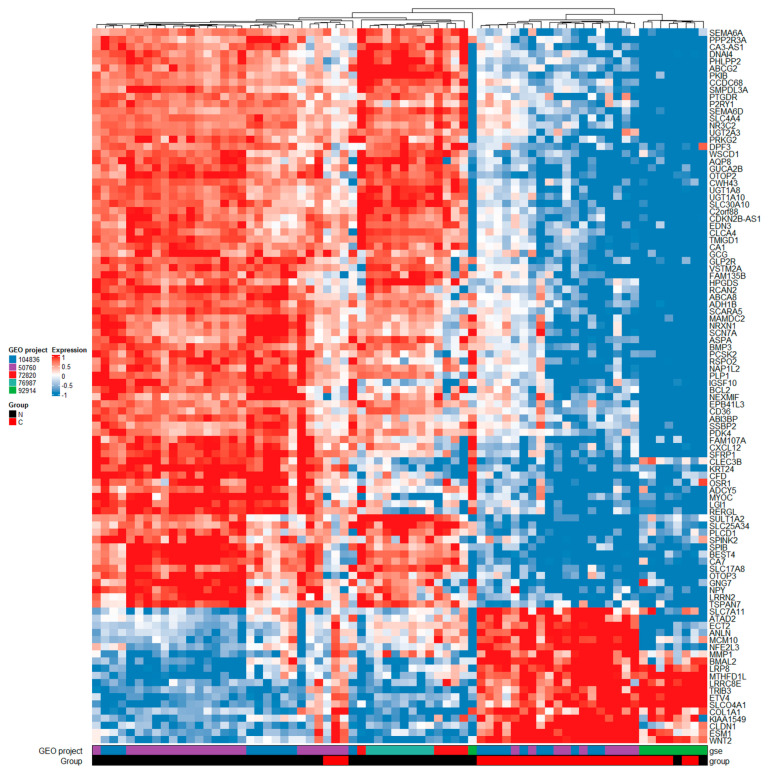
Heatmap of top 100 DEGs between normal tissue and carcinomas. Legend: GEO, gene expression omnibus; GSE, gene expression omnibus series; group N, normal tissue; group C, carcinoma tissue; gene names are shown on the right side of the heatmap; downregulation is associated with blue color, no difference in expression is associated with white color, and upregulation is associated with red color.

**Figure 3 ijms-23-13252-f003:**
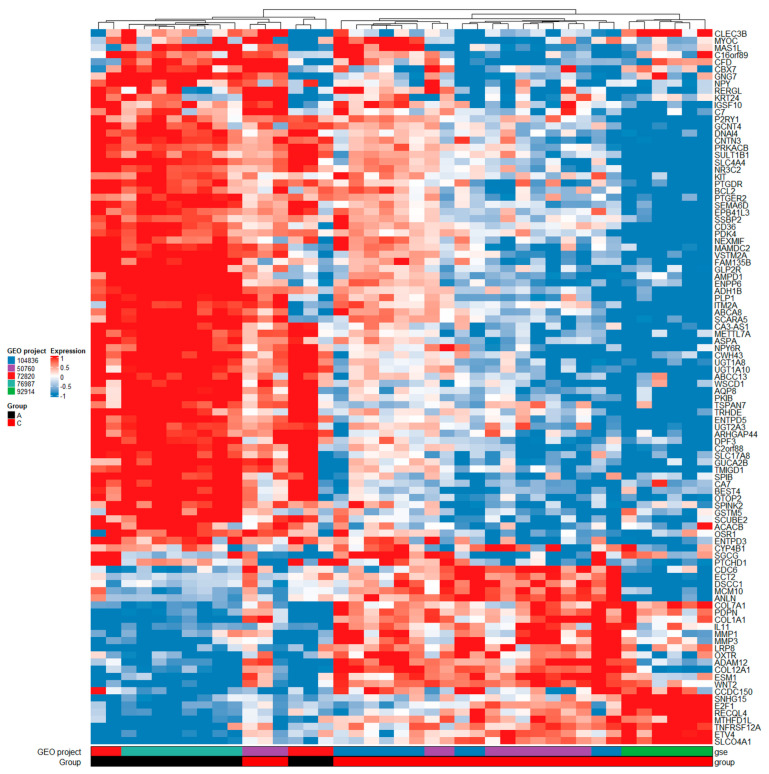
Heatmap of top 100 DEGs between adenomas and carcinomas. Legend: GEO, gene expression omnibus; GSE, gene expression omnibus series; group A, adenoma; group C, carcinoma tissue; gene names are shown on the right side of the heatmap; downregulation is associated with blue color, no difference in expression is associated with white color, and upregulation is associated with red color.

**Figure 4 ijms-23-13252-f004:**
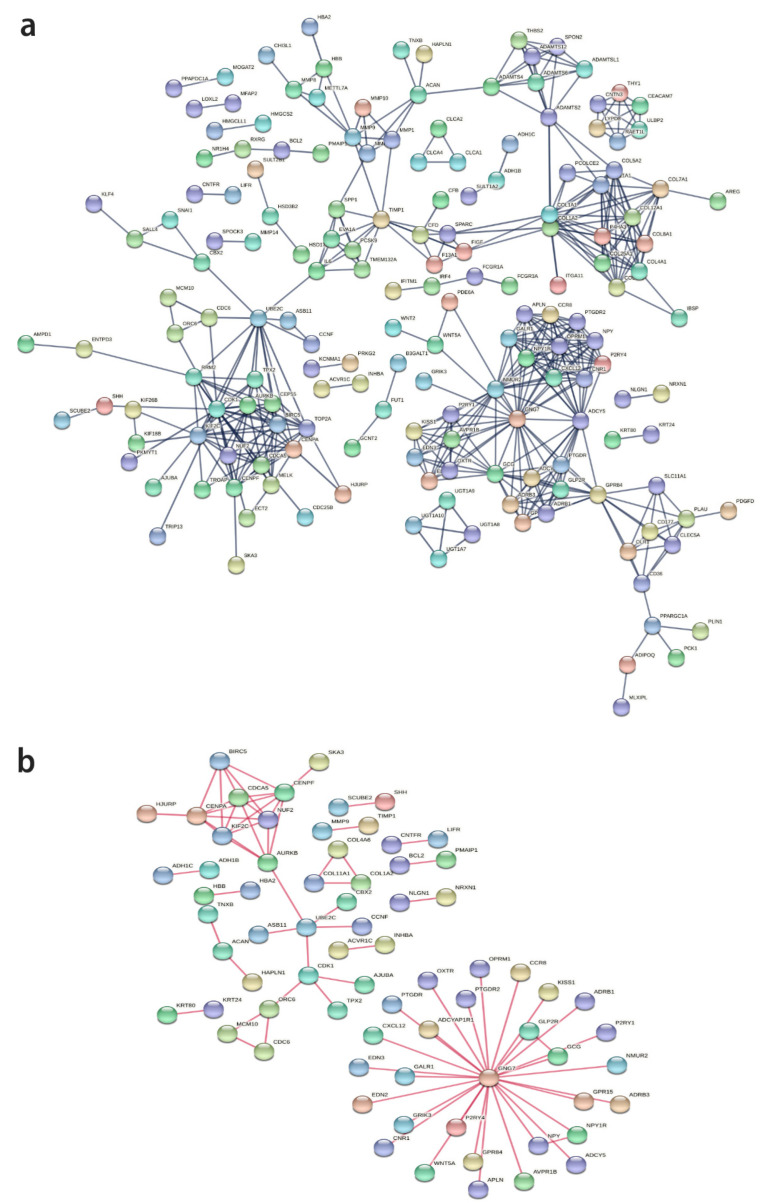
Protein interaction network: (**a**) full protein–protein interaction network; (**b**) physical protein–protein interaction network. Legend: nodes are associated with proteins; edges are associated with protein–protein interactions.

**Figure 5 ijms-23-13252-f005:**
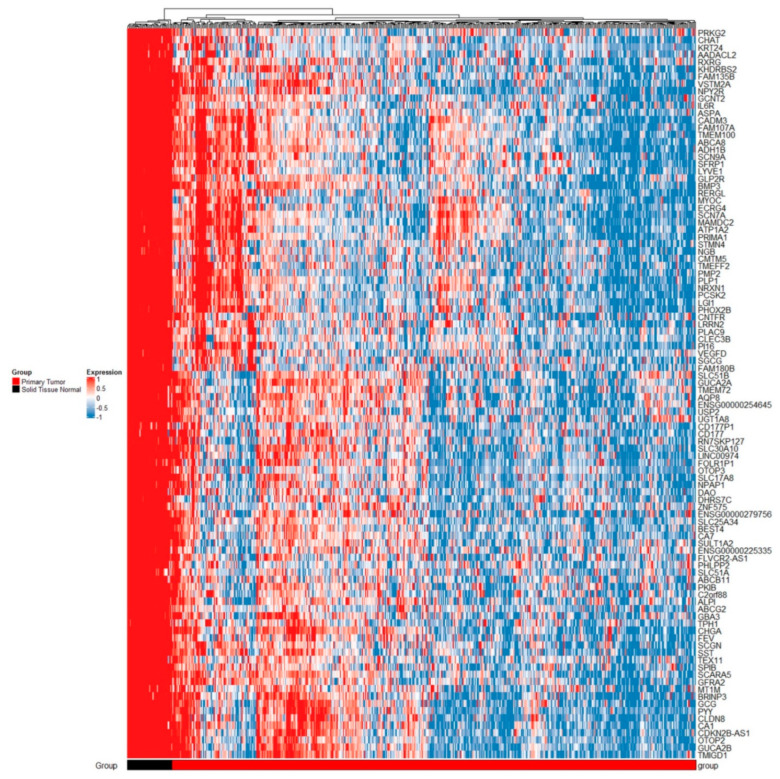
Top 100 DEGs heatmap of expression between the primary tumor and normal samples. Legend: group Primary Tumor, carcinoma; group Solid Tissue Normal, normal tissue; gene names, where available, otherwise Ensembl Gene identificators are shown on the right side of the heatmap; downregulation is associated with blue color, no difference in expression is associated with white color, and upregulation is associated with red color.

**Figure 6 ijms-23-13252-f006:**
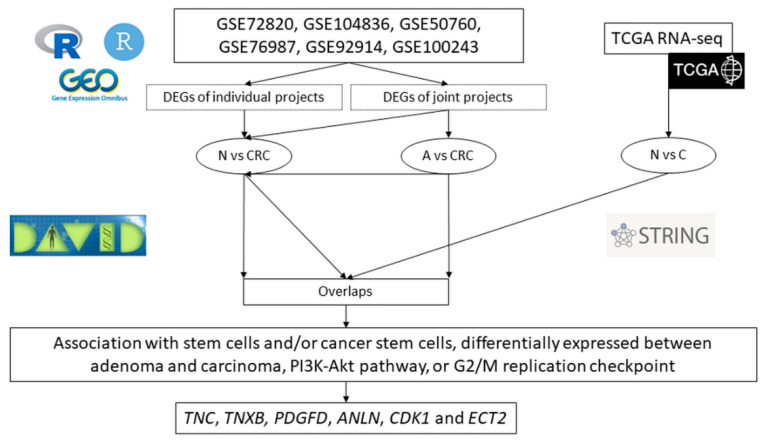
Workflow of identification of differentially expressed genes in the bioinformatics analysis. Legend: N, normal colon tissue; A, adenomas; CRC, colorectal cancer; DEGs, differentially expressed genes; TCGA, the cancer genome atlas; R, R programming language; R studio, R studio programme; DAVID, The Database for Annotation, Visualization and Integrated Discovery; STRING; STRING database for protein–protein interaction analysis.

**Figure 7 ijms-23-13252-f007:**
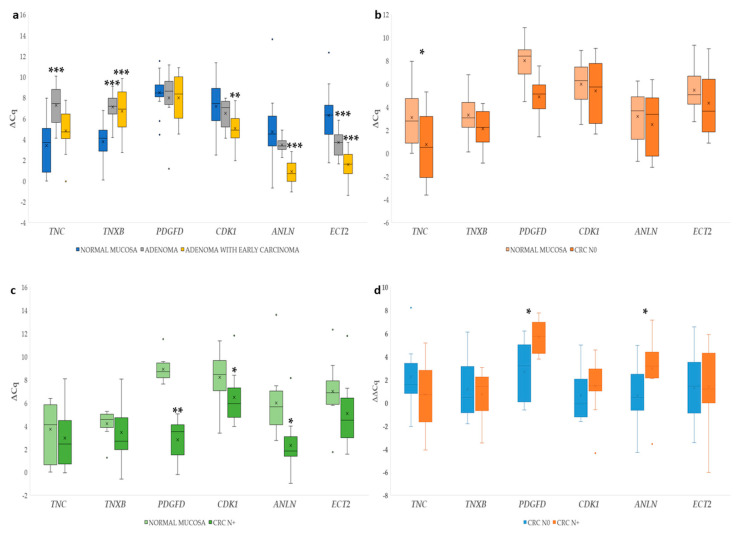
Expression of six genes (*TNC*, *TNXB*, *PDGFD*, *CDK1*, *ANLN*, *ECT2*) in colorectal cancerogenesis: (**a**) ∆Cq of six genes in normal mucosa, adenoma, and adenoma with early carcinoma; (**b**) ∆Cq of six genes in colorectal cancer without lymph node metastases (CRC N0) compared to corresponding normal mucosa; (**c**) ∆Cq of six genes in colorectal cancer with lymph node metastases (CRC N+) compared to corresponding normal mucosa; (**d**) ∆∆Cq of six genes in colorectal cancer with (CRC N+) compared to those without lymph node metastases (CRC N0). Legend: x, mean; ◦, outlier; *, *p* ≤ 0.05; **, *p* ≤ 0.01; ***, *p* ≤ 0.001.

**Table 1 ijms-23-13252-t001:** Results of identification of differentially expressed genes (DEGs) on individual projects, including project number, comparison, number of DEGs, and additional data.

Project Number	Comparison	Number of DEGs	Additional Data
**GSE72820**	7 adenomas and corresponding normal tissue	/	/
**GSE76987**	8 adenomas and corresponding normal tissue	3 *	Appendix A, table GSE76987
**GSE104836**	10 carcinomas and corresponding normal tissue	1544	Appendix A, table GSE104836
**GSE92914**	3 carcinomas and corresponding normal tissue	/	/
**GSE100243**	6 carcinomas and corresponding normal tissue	2470	Appendix A, table GSE100243
**GSE50760**	18 carcinomas and corresponding normal tissue	1766	Appendix A, table GSE50760

Legend: * the procedure for identification of DEGs in this project resulted in 3 DEGs at logarithm of base 2 of the fold change value (logFC) > 1.5 and *p* < 0.05, with “no adjustment” setting used in the toptable function for comparing paired normal tissue and adenomas. With standard settings, no DEGs were identified.

**Table 2 ijms-23-13252-t002:** All statistical comparisons for the investigated genes.

*TNC*	Normal Mucosa	Adenoma	Adenoma with Early Carcinoma	CRC N0	CRC N+
Normal mucosa	1	0.000	/	0.039	/
Adenoma		1	0.012	0.000	0.002
Adenoma with early carcinoma			1	0.003	/
CRC N0				1	/
CRC N+					1
** *TNXB* **	**Normal mucosa**	**Adenoma**	**Adenoma with early carcinoma**	**CRC N0**	**CRC N+**
Normal mucosa	1	0.000	0.000	/	/
Adenoma		1	/	0.000	0.001
Adenoma with early carcinoma			1	0.000	0.004
CRC N0				1	/
CRC N+					1
** *PDGFD* **	**Normal mucosa**	**Adenoma**	**Adenoma with early carcinoma**	**CRC N0**	**CRC N+**
Normal mucosa	1	/	/	/	0.008
Adenoma		1	/	0.018	0.003
Adenoma with early carcinoma			1	0.029	0.000
CRC N0				1	0.03
CRC N+					1
** *CDK1* **	**Normal mucosa**	**Adenoma**	**Adenoma with early carcinoma**	**CRC N0**	**CRC N+**
Normal mucosa	1	/	0.006	/	0.042
Adenoma		1	0.027	/	/
Adenoma with early carcinoma			1	/	/
CRC N0				1	/
CRC N+					1
** *ANLN* **	**Normal mucosa**	**Adenoma**	**Adenoma with early carcinoma**	**CRC N0**	**CRC N+**
Normal mucosa	1	/	0.000	/	0.019
Adenoma		1	0.000	/	/
Adenoma with early carcinoma			1	/	/
CRC N0				1	0.046
CRC N+					1
** *ECT2* **	**Normal mucosa**	**Adenoma**	**Adenoma with early carcinoma**	**CRC N0**	**CRC N+**
Normal mucosa	1	0.000	0.000	/	/
Adenoma		1	0.002	/	0.02
Adenoma with early carcinoma			1	0.023	0.001
CRC N0				1	/
CRC N+					1

Legend: CRC N0, colorectal cancer without lymph node metastases; CRC N+, colorectal cancer with lymph node metastases; /, not statistically significant.

**Table 3 ijms-23-13252-t003:** Overview of projects included in the study.

Project	Platform	Tissue type	Normal	Adenoma	Carcinoma
GSE72820	Illumina	No data	7	7	0
GSE104836	Illumina	Fresh frozen	10	0	10
GSE50760	Illumina	No data	18	0	18
GSE76987	Illumina	RNA-later	8	8	0
GSE92914	Illumina	Fresh frozen	3	0	3
GSE100243	Ion Torrent	FFPE	6	0	8

**Table 4 ijms-23-13252-t004:** Characteristics of the included patients.

Patients	Adenoma	Adenoma with Early Carcinoma	CRC without Lymph Node Metastases	CRC with Lymph Node Metastases
M:F	10:1	9:4	4:6	9:4
Age	62.3 ± 10.7	64.9 ± 5.7	72.7 ± 11.6	73.2 ± 11.8

Legend: CRC, colorectal cancer; F, female; M, male.

**Table 5 ijms-23-13252-t005:** Selected probes.

Gene	Entrez Gene ID	Assay ID
** *B2M* **	567	Hs99999907_m1
** *IPO8* **	10526	Hs00183533_m1
*TNC*	3371	Hs01115665_m1
*TNXB*	7148	Hs00372889_g1
*PDGFD*	80310	Hs00228671_m1
*CDK1*	983	Hs00364293_m1
*ANLN*	54443	Hs01122612_m1
*ECT2*	1894	Hs00978168_m1

## Data Availability

All scripts and ∆∆Cq data are available upon reasonable request from the authors.

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
