# Peer review of "Bioinformatics Analysis of RNA-seq Data Reveals Genes Related to Cancer Stem Cells in Colorectal Cancerogenesis"

_ijms, 2022, doi:10.3390/ijms232113252_

Round 1
Reviewer 1 Report
Comments ijms-1966512
Manuscript ijms-1966512 is a bioinformatic work studying genes related to cancer stem cells in colorectal cancerogenesis from public RNA-seq data. The authors then preformed their own experiment to validate the identified DEGs by qPCR analysis using several selected genes. However, major and minor issues are found in the text that need to be resolved before the ms is suitable for publication. I have a concern how the authors build the paragraph in the text where some of them only consist of 1 or two sentences. Those paragraphs could be expanded or even combined with others. At least three sentences are needed to build a paragraph.
Abstract
17-8 - rewrite the sentence!
Introduction
60 - should be “genes expression”
Results
79 - “differentially expressed genes” can be omitted!
81 - What do PEI and PPI stand for? State them!
83 - number of DEGs
92 & 113 - “that were differentially expressed” can be omitted!
97 - put the project number at the appropriate spot on the figure!
107, 129, 156, & 207 - “using STRING” can be omitted!
107-9, 129-31, & 207-8 - it would be more informative to present the PPI analysis also using a Figure, like Figures 5 & 6.
123, 149, & 201 - “the Gene ontology” can be omitted!
135-6 - “and analyzed …… section” can be omitted!
138 & 141 - are these p values after adjustment? See also line 184
162 & 167 - the name of DEGs should be included on the figure of heat map (Figures 3 & 4)!
173 & 176 - the legend of figures 5 & 6 is not complete! What is the different between both figures?
187-9 - Alternatively, PPI could be conducted using the selected DEGs, for example: top 100 DEGs
191 - see the previous comment on lines 162 & 167!
225-33 - it would be better to send this subsection to M&M section!
Discussion
310-3 - the complete name of each gene mentioned here should be included!
349 - degree of mRNA level
- conclusions section needs to be included in the ms!
Materials and methods
570 - “mRNAs of” can be omitted!
574 - Table 5 is incomplete. Some necessary information should be included such as sequences of primers and probes, acc. Number of each gene, target size of each pair primer, etc. Reference genes are not bold on the figure.
583 - state the concentration of primers, probes, and cDNA applied!
588 - was it 2-△△Cq?? The authors could refer to Livak and Schmittgen 2001!
590 - see comment on line 588!
597 - mRNAs or genes??

Reviewer 2 Report
1, Kristian et al. have identified and verified six CSC related genes, which may participate in the early stage of CRC development and subsequent transfer process. The results provide further evidence for the involvement of CSC related genes in the development and progression of CRC.So, are only the six CSC genes related to CRC development screened, or are they finally determined according to some criteria? Are there detailed research data on the differential expression of these six genes in CRC and other cancer cells?
2, In the study of protein interaction (PPI), the author established a physical protein interaction network. During the analysis, the author should clarify whether these genes directly interact with other proteins or whether the expression products of these genes interact with other ligands.
3, Using qPCR and tissue samples from 47 patients with adenoma, adenoma with early cancer, cancer without lymph node metastasis and cancer with lymph node metastasis, and comparing with normal mucosa, we verified the identified CSC related genes. Is the sample size selected relatively small? How to eliminate the adverse effects of small samples on the reliability and accuracy of research results? The author should analyze this.
4, By identifying the positive correlation between PDGFD, ANLN, CDK1 and ECT2 and the degree of malignancy, the authors suggest that these genes should be considered as potential biomarkers of CRC progression. However, these genes are basically related to DNA replication and cell division, and have significant differences in expression in most tumor cells and at different stages of tumor cell progression. They are not specific in the expression of certain tumor stem cells. Therefore, is it reasonable to regard these genes as potential CRC related biomarkers for malignant transformation, progression and metastasis of CRC?
5, Figures 1 and Figures 2 should be combined into one figure.Figures 5 and Figures 6 should be combined into one figure. Figure 8, Figure 9, Figure 10 and Figure 11 should also be combined into one figure.
6,Minor issues:
P9, line 202 : localisation ==> localization
p10, line 233 : Table 3. ==> Table 2.
p16, line 362 :long term ==> long-term
Round 2
Reviewer 1 Report
Comments ijms-1966512-v2
The authors have significantly improved the quality of manuscript ijms-1966512 according to the previous comments. However some minor issues still exist in the revised ms. This ms could be considered for publication after addressing the remaining issues as follows:
142 & 145 - it would be simpler by just saying adj p<0.05!
165, 171 & 195 - it would be more informative to include the name of genes (the annotated genes) instead of their IDs!
446 - should be “Table 3”
562 - should be “Table 4”
604 - Technical question: were the primers combined with the Taqman probe? I did not see any primers added in the qPCR mix reactions!
